# Morphological and Molecular Description of *Sarcocystis* *myodes* n. sp. from the Bank Vole (*Clethrionomys glareolus*) in Lithuania

**DOI:** 10.3390/biology11040512

**Published:** 2022-03-26

**Authors:** Eglė Rudaitytė-Lukošienė, Marius Jasiulionis, Linas Balčiauskas, Petras Prakas, Vitalijus Stirkė, Dalius Butkauskas

**Affiliations:** Nature Research Centre, Akademijos 2, LT-08412 Vilnius, Lithuania; egle.rudaityte@gamtc.lt (E.R.-L.); marius.jasiulionis@gamtc.lt (M.J.); linas.balciauskas@gamtc.lt (L.B.); petras.prakas@gamtc.lt (P.P.); vitalijus.stirke@gamtc.lt (V.S.)

**Keywords:** bank vole, *Sarcocystis myodes*, species description, microscopy, molecular characterization, phylogeny

## Abstract

**Simple Summary:**

Representatives of genus *Sarcocystis* (Apicomplexa, Sarcocystidae) are parasites of mammals, birds, and reptiles. They are characterized by two-host prey-predator life cycle. Rodents are reservoirs of zoonotic diseases and play a significant role in a spread of pathogens. At present, about 40 *Sarcocystis* species are known to form sarcocysts in muscles and brain of rodents. Most of *Sarcocystis* spp. in these hosts have been characterized by morphological methods and life cycle investigations. In the present study a new *Sarcocystis* species, *S*. *myodes* is described in skeletal muscles of the bank voles (*Clethrionomys glareolus*) from Lithuania using morphological, molecular, and phylogenetic analysis. Based on five genetic loci, *S*. *myodes* was most closely related to *Sarcocystis* spp. using predatory mammals as their definitive hosts. The analysis of previous studies indicates that *Sarcocystis* spp. diversity in voles is not fully revealed. Furthermore, some *Sarcocystis* spp. formerly detected in voles are important pathogens. Therefore, further molecular examinations are needed for the revision of *Sarcocystis* spp. in these hosts.

**Abstract:**

Numerous rodent species have been broadly examined for *Sarcocystis* parasites. Nevertheless, recent investigations on *Sarcocystis* spp. in voles are lacking. As many as 45 bank voles (*Clethrionomys* *glareolus*) captured in several locations in Lithuania were examined in the present study. Based on morphological, genetic, and phylogenetic results, sarcocysts detected in one bank vole were described as *Sarcocystis myodes* n. sp. Using light microscopy analysis, the observed sarcocysts were ribbon-shaped, 6000–3000 × 70–220 µm in size. Sarcocysts were characterized by a relatively thin (about 1 μm) and apparently smooth cyst wall. The lancet-shaped bradyzoites were 9.6–12.0 × 3.1–4.6 μm in size. By transmission electron microscopy, the sarcocyst wall was up to 1 μm thick, parasitophorous vacuolar membrane had small knob-like blebs. Based on *18S* rDNA, *28S* rDNA, *cox1*, *rpoB*, and *ITS1* loci, *S*. *myodes* showed highest similarity with *S*. *ratti* from the black rat (*Rattus rattus*). According to phylogenetic placement, *S*. *myodes* was most closely related to *Sarcocystis* spp. that employ predatory mammals as their definitive hosts. Morphologically, sarcocysts of *S*. *myodes* have similar features to those of *S*. *cernae*, *S*. *dirumpens*, and *S*. *montanaensis* described in voles, however, they use birds of prey or snakes as their definitive hosts.

## 1. Introduction

*Sarcocystis* parasites belonging to the order Apicomplexa and Sarcocystidae family are worldwide distributed protozoan parasites, having two hosts in their life cycle. More than 200 species of *Sarcocystis* species have been described in reptiles, birds, and mammals. The intermediate host becomes infected by ingesting sporocyst-contaminated food or water, while the definitive host gets infected by eating muscular or neural tissues containing mature sarcocysts [1].

Rodentia is the most abundant group of mammals [2]. These animals are important components of the food chain [2,3,4,5] and play a significant role in the spread of pathogens due to their widespread distribution [6,7]. Currently, approximately 40 *Sarcocystis* species are known to infect rodents [1]. The majority of these species have been described on the basis of morphological and life cycle studies [1]. Only 10 of them, *S*. *cymruensis*, *S*. *muris*, *S*. *ratti*, *S*. *dispersa*, *S*. *glareoli*, *S*. *microti*, *S*. *atheridis*, *S*. *singaporensis*, *S*. *zamani* and *S*. *zuoi* have been molecularly characterized [8,9,10,11,12,13,14,15]. It should be noted that mice and rats were examined for *Sarcocystis* infection most thoroughly [1].

The bank vole *Clethrionomys* (*Myodes*) *glareolus* is one of the most common vole species in forests, shrubby habitats, fruit orchards, and commensal habitats of Lithuania [16,17,18,19]. At present, five species of *Sarcocystis* are known in bank voles. Three of them are described in the muscles of European voles (*S*. *clethrionomyelaphis*, *S*. *dirumpens*, and *S*. *muriviperae*) and are transmitted via snakes [20,21,22], while two species, *S*. *glareoli* and *S*. *microti* (previously assigned to genus *Frenkelia*), are described in the brain of rodents and use buzzards (*Buteo* spp.) as their definitive hosts [10,23,24]. Among these species, *S*. *glareoli* and *S*. *microti* have been characterized in complete *28S* rDNA (~3280 bp) and partial *18S* rDNA (~1630 bp). In the present paper, we describe a new species of *Sarcocystis* detected in the bank vole from Lithuania by microscopical and DNA sequence analysis.

## 2. Materials and Methods

### 2.1. Collection of Samples

We investigated 45 bank voles, trapped in 2018 at three great cormorant colonies in Lithuania, located in western (*n* = 4), southern (*n* = 8), and eastern (*n* = 33) parts of the country (Figure 1). Only adults were selected for the analysis because of their increased risk of infection. A portion of skeletal muscles were stored at −20 °C to examine for *Sarcocystis* infection.

### 2.2. Morphological Examination

To detect sarcocysts, fragments of muscle tissue (~0.5–1 g) were stained with methylene blue solution as described by Prakas et al. [15] and studied under light microscopy (LM). Morphological analysis of the observed sarcocysts was performed in fresh-squashed *Sarcocystis*-positive muscle samples. Sarcocysts were isolated with fine preparation needles. The excised sarcocysts were morphologically characterized on the basis of the form and size of cysts and bradyzoites released from the cyst, as well as the structure of the cyst wall. Three sarcocysts from fresh muscle preparations of the only one infected individual of the bank vole were isolated, and stored for further DNA extraction in sterile 1.5 mL microcentrifuge tubes with 96% ethanol. Sarcocyst wall ultrastructure was examined by transmission electron microscopy (TEM) using the previously described method [15].

### 2.3. DNA Extraction and PCR

Total DNA extraction procedure was conducted by a commercial kit “GeneJet Genomic DNA Purification Kit” (Thermo Fisher Scientific, Vilnius, Lithuania). The isolated sarcocysts were characterized molecularly in five genetic markers, three nuclear loci (*18S* rDNA, *28S* rDNA, and *ITS1*), one mitochondrial gene (*cox1*), and one apicoplastic gene (*rpoB*). The primer pairs SarAF/SarBR and SarCF/SarDR (for nearly complete *18S* rDNA), KL-P1F/KL-P2R (for partial *28S* rDNA), P-ITSF/P-ITSR (for complete *ITS1* region), SF1/SR5 (for partial *cox1*), and RPObF/RPObR (for partial *rpoB*) were used for amplification [25,26,27]. DreamTaq PCR Master Mix (Thermo Fisher Scientific, Vilnius, Lithuania) was used to perform PCR in the final 25 μL volume according to the manufacturer’s instructions. The PCR cycling conditions were as follows: an initial hot start for 5 min at 95 °C, followed by 35 cycles of denaturation for 45 s at 94 °C, annealing for 60 s at 50–58 °C depending on the primer pair, elongation for 70 s at 72 °C, and final extension for 7 min at 72 °C [28].

### 2.4. Sequence Analysis

The amplified PCR fragments were evaluated visually using 1% agarose gel electrophoresis. PCR products of *18S* rDNA, *28S* rDNA, and *rpoB* regions were purified with the help of ExoI and FastAP (Thermo Fisher Scientific Baltics, Vilnius, Lithuania) and sequenced directly using the same forward and reverse primers as for PCR. Sequencing reactions were carried out according to the manufacturer’s instructions using the Big-Dye^®^ Terminator v3.1 Cycle Sequencing Kit (Thermo Fisher Scientific, Vilnius, Lithuania) and the 3500 Genetic Analyzer (Applied Biosystems, Foster City, CA, USA). The *18S* rDNA gene region sequences were obtained by amplification and sequencing with two pairs of primers, with the resulting sequences manually edited and merged into a single fragment. Samples amplified by means of the primers intended for *cox1* and *ITS1* regions gave unspecific products, thus before sequencing they were extracted from the gel and purified using commercial kit “GeneJET Gel Extraction Kit” (Thermo Fisher Scientific, Vilnius, Lithuania). In the current study, obtained sequences were analyzed using online Nucleotide BLAST program (http://blast.ncbi.nlm.nih.gov/, accessed on 17 January 2022). Conducting phylogenetic analysis, the sequences gained in the present study were compared with the very similar sequences of numerous *Sarcocystis* species. For this purpose, the sequence alignments were generated with the MUSCLE algorithm available in MEGAX software [29]. Phylogenetic trees were obtained using Bayesian method implemented in TOPALi software [30]. *Toxoplasma gondii* was set as the outgroup for the examined *Sarcocystis* species in all phylogenetic analyses.

## 3. Results

### 3.1. Prevalence and Morphological Description of Sarcocysts

Using methylene-blue staining, sarcocysts were detected in one muscle sample of one bank vole out of 45 voles examined (2.2% prevalence; 95% CI (confidence interval) = 0.4–11.6%). The infected individual (N26), a female from the bank of the Nemunas River in South Lithuania (Figure 1), was found to have 36 sarcocysts in 0.5 g of muscle.

Three sarcocysts, morphologically varying in size, were isolated from the infected muscle sample. All sarcocysts had a smooth cyst wall with no visible protrusions (Figure 2a) and further molecular analysis showed that they belonged to the same species. Sarcocysts were ribbon-shaped and measured 1970 × 150 µm (600–3000 × 70–220 µm; *n* = 5). The cyst wall was thin (~1 μm) and smooth. The septa were clearly visible. They divided sarcocysts into chambers containing numerous lancet-shaped 10.9 × 3.9 (9.6–12.0 × 3.1–4.6; *n* = 12) μm bradyzoites (Figure 2b).

The TEM analysis showed that cyst wall was up to 1 μm in thickness. Parasitophorous vacuolar membrane had small knob-like blebs and was slightly wavy where the ground substance layer passed inside the cyst as septa (Figure 2c,d). The ground substance was smooth. The cyst wall ultrastructure was similar to type 1a described by Dubey et al. [1].

### 3.2. Molecular Characterization and Phylogeny

Three sarcocysts were characterized at five genetic loci and the molecular data obtained indicated that all three isolates belong to the same species. The resulting sequences were identical to each other at four loci (*18S* rDNA, *28S* rDNA, *cox1*, and *rpoB*), and two different sequences with 5 SNPs were obtained in the *ITS1* region. The obtained sequences had highest similarity values to those of *S*. *ratti* described in muscles of the black rat (*Rattus rattus*), except when comparing *rpoB* sequences (Table 1). Based on *rpoB*, the analyzed sequences were most similar to those of *S*. *fulicae* from the Eurasian coot (*Fulica atra*), *S*. *cornixi* from hooded crow (*Corvus cornix*), *S*. *caninum* from dog, and *S*. *arctica* from red fox (*Vulpes vulpes*). It should be noted that *rpoB* sequences have not been determined for *S*. *ratti* yet. The greatest differences with the closely related *Sarcocystis* species were detected in the *ITS1* region (>20%), moderate differences were obtained in *rpoB* and *28S* rDNA (~2–4%), while *cox1* and *18S* rDNA sequences of *Sarcocystis* sp. from the bank vole differed by less than 1% as compared with those of *S*. *ratti*. Except for *S*. *ratti*, the *ITS1* sequences of *S*. *myodes* were highly distinct from and did not significantly match those of other *Sarcocystis* species available in GenBank. As a result, no additional molecular analysis of *ITS1* sequences was performed.

Four different molecular loci were used to build phylogenetic trees. The Hasegawa–Kishino–Yano (HKY) model was used to construct phylogenetic trees for *18S* rDNA, *28S* rDNA, and *rpoB*, while the generalized time-reversible (GTR) model was chosen as the evolutionary model for *cox1* tree construction. Currently, of the *Sarcocystis* species whose intermediate hosts are rodents, the most readily available are the *18S* rDNA sequences. The final *18S* rDNA sequence alignment used 35 sequences and 1652 aligned nucleotide positions. The phylogenetic analysis based on *28S* rDNA sequences included 32 sequences and 1347 aligned nucleotide positions. For *cox1* and *rpoB* phylogenetic analysis, 24 sequences with 976 aligned nucleotide positions and 19 sequences with 694 aligned nucleotide positions were used, respectively. In the phylogenetic trees constructed using *18S* rDNA, *28S* rDNA, and *cox1* sequences (Figure 3a–c), *Sarcocystis* sp. from the bank vole was a sister taxon to *S*. *ratti* and showed a close relationship with *S*. *cymruensis* and *S*. *muris* using the brown rat (*Rattus norvegicus*) and the house mice (*Mus musculus*) as their intermediate hosts, respectively [14,31]. Two of these species, *S*. *cymruensis* and *S*. *muris*, employ predatory mammals as their definitive hosts [14,31,32], while the final hosts of *S*. *ratti* are unknown. Based on *18S* rDNA, *Sarcocystis* spp. employing rodents as intermediate hosts and snakes as definitive hosts were placed into one cluster, while the phylogenetic position of the other *Sarcocystis* spp. that form sarcocysts in the muscles of rodents and employ birds or mammals as definitive hosts was not clear. In the phylograms of *28S* rDNA and *cox1* sequences, the *Sarcocystis* sp. from the bank vole did not form a single clade with *S*. *glareoli*, *S*. *jamaicensis*, *S*. *microti*, and *S*. *strixi* characterized by rodent-bird (intermediate-definitive host) life cycle [23,33,34]. The *Sarcocystis* sp. from the bank vole formed a separate branch in the phylogram constructed using *rpoB* sequences (Figure 3d). However, it should be taken into account that the *rpoB* sequences of *S*. *ratti*, *S*. *cymruensis*, and *S*. *muris*, most closely related to the *Sarcocystis* sp. from the bank vole, are not available in the GenBank.

### 3.3. Description of Sarcocystis myodes n. sp.

The *Sarcocystis* parasite discovered in the present study was morphologically distinct from the three species known to inhabit the musculature of bank voles, *S*. *clethrionomyelaphis*, *S*. *dirumpens*, and *S*. *muriviperae*. Sarcocysts detected in the bank vole from Lithuania were morphologically similar to those of *S*. *glareoli* and *S*. *microti*, however, the latter two species are confined to the brain. Additionally, reliable genetic differences were distinguished when comparing *S*. *glareoli* and *S*. *microti* with the species examined in the present study (2.3–2.4% in *18S* rDNA and 6.6–7.1% in *28S* rDNA). Other *Sarcocystis* species detected in rodents of the Cricetidae family that are morphologically similar to the species observed in the bank vole in this study are transmitted by birds or snakes. Phylogenetic analysis of the present work shows that *Sarcocystis* sp. from the Lithuanian bank vole does not group with *Sarcocystis* species such as *S*. *dispersa*, *S*. *glareoli*, *S*. *jamaicensis*, *S*. *microti*, and *S*. *strixi* that use birds as definitive hosts [23,33,34,35] or such species as *S*. *singaporensis* and *S*. *atheridis* that use snakes as definitive hosts [36,37]. When comparing the morphology of the *Sarcocystis* sp. from bank vole observed in the present study with that of the most closely related species, *S*. *ratti*, both parasites have a very similar sarcocyst size, shape, and cyst wall structure, but the species differ in the morphometric parameters of the bradyzoites. It was observed that the bradyzoites of the *Sarcocystis* sp. from the bank vole (9.6–12.0×3.1–4.6 µm) are longer than those of *S*. *ratti* (7.5–9.3 × 3.9–4.8 µm). Taking into account sarcocyst morphology and genetic data obtained, a new species, *Sarcocystis myodes*, is proposed for an organism found in the muscles of the bank vole form Lithuania.

Taxonomic summary of *S*. *myodes* n. sp.

Type intermediate host: The bank vole *Clethrionomys* (*Myodes*) *glareolus*.

Definitive host: Unknown.

Locality: Lithuania.

Morphology of sarcocyst: By LM, sarcocysts were microscopic, ribbon-shaped, 600–3000 × 70–220 µm in size, having a thin (~1 µm) and smooth cyst wall without visible protrusions. Lancet-shaped bradyzoites measured 9.6–12.0 × 3.1–4.6 μm. By TEM, knob-like blebs covered the parasitophorous vacuolar membrane. Type 1a-like.

Specimens deposited: TEM material is available at Laboratory of Molecular Ecology of the Nature Research Centre, Vilnius, Lithuania. All sequences obtained in the study are deposited in NCBI GenBank with accession numbers OM523014–OM523016 (*18S* rDNA), OM523017–OM523019 (*28S* rDNA), OM486937–OM486939 (*cox1*), OM486940–OM486942 (*rpoB*), and OM523020–OM523022 (*ITS1*).

Etymology: The Latin name of genus *Myodes* is used for the species name.

Recorded in URN as urn:lsid:zoobank.org:act:427763E1-B75E-40F7-8CF9-5079C6895459.

## 4. Discussion

The prevalence of *Sarcocystis* infection in the bank vole obtained in the present study (2.2%) is similar to that (1.8–14.4%) detected in this host in the previous studies carried out in Lithuania [38,39,40,41]. There is no information in other countries on the prevalence of *Sarcocystis* muscular infection in naturally infected bank voles. In general, *Sarcocystis* infection rates in various rodent species from different countries are relatively low/moderate. For instance, the prevalence of *Sarcocystis* spp. was 0.7% in synanthropic rodents from Spain [42], 11.6% in wild rats (*Rattus* spp.), 25% in large oriental voles (*Eothenomys miletus*) in China [43,44], 15.4% in black rats from Latvia [15], 33% in wild rodents from Thailand [45], and 40% in wild rodents from Indonesia [46]. Based on morphological *Sarcocystis* spp. studies conducted into rodents from Lithuania, it can be concluded that infection prevalence depends on the host species [38,39,40,41]. The lowest infection rates (up to 6%) were previously reported in mice species, the yellow-necked mouse (*Apodemus flavicollis*), the striped field mouse (*Apodemus agrarius*), and the domestic mouse. In five examined vole species (European water vole, short-tailed field vole, common vole, tundra vole, and bank vole), *Sarcocystis* infection prevalence ranged from 1.8% to 20.0%, whereas the highest *Sarcocystis* infection rates (30.2–52.4%) were detected in rats, the brown rat, and the black rat.

The DNA sequence comparison at five loci revealed significant genetic differences between *S*. *myodes* and other *Sarcocystis* spp., using rodents as their intermediate host. At the *18S* rDNA, *cox1*, *28S* rDNA, and *ITS1* regions, differences between *S*. *myodes* and the most closely related species, *S*. *ratti*, were 0.5%, 0.6%, 1.9%, and 22.7–23.0%, respectively. Phylogenetic analysis showed that *S*. *myodes* had no close relationship with *Sarcocystis* spp. employing rodents as their intermediate hosts and birds or snakes as their definitive hosts, and that *S*. *myodes* was placed together with *S*. *ratti* and *S*. *cymruensis* (Figure 3). Rats are intermediate hosts to both *S*. *ratti* and *S*. *cymruensis* [47]. Domestic cats have been shown to be definitive hosts of *S*. *cymruensis* [14], and in some cases, due to cannibalism, rats may serve not only as an intermediate but also as a definitive host of the aforementioned species [43]. Therefore, phylogenetic evidence suggests that predatory mammals are most likely definitive hosts of *S*. *myodes*. The studied area abounds in foxes, raccoon dogs, minks, and martens [48], all of which feed on rodents, including the bank vole [16].

To date, three *Sarcocystis* species have been identified in the muscles of bank voles, and all of them being transmitted by snakes. Sarcocysts of *S*. *clethrionomyelaphis* and *S*. *muriviperae* have distinct wall structures (about 3–3.5 μm protrusions) that differ from the sarcocysts described in this work, whereas *S*. *dirumpens* has an ultrastructure similar to that of *S*. *myodes*. The sarcocysts wall of *S*. *dirumpens* was thin, corrugated, or folded, and the primary cyst wall was evaginated with knob-like blebs [49]. However, bradyzoites of *S*. *dirumpens* were smaller (7.0–9.0 × 1.6–2.0 µm) [20,49] as compared to those of *S*. *myodes* (9.6–12.0 × 3.1–4.6 μm). Sarcocysts that were morphologically similar to those of *S*. *myodes* were previously reported in bank voles from Lithuania [39]. Based on LM, sarcocysts had thin walls (0.7–1.4 µm) without visible protrusions and relatively large (10.0–14.2 × 2.3–4.7 µm; *n* = 231) bradyzoites. In the same study, sarcocysts morphologically similar those of *S*. *myodes* were also described in short-tailed field vole. *Sarcocystis* spp. morphologically similar to *S*. *myodes* were found in other vole species. Based on the structure of the sarcocyst wall, *S*. *myodes* is similar to *S*. *cernae* from the common vole and *S*. *montanaensis* from the eastern meadow vole (*Microtus pennsylvanicus*), the long-tailed vole (*Microtus longicaudus*), and the prairie vole (*Microtus achrogaster*) [50,51,52]. Transmission experiments revealed that the common kestrel (*Falco tinnunculus*) was a definitive host of *S*. *cernae* [50]. Intermediate hosts of *S*. *montanaensis* are prevalent in North America, and it has been demonstrated that this parasite uses a variety of snake species as definitive hosts [52,53]. Bradyzoites of *S*. *cernae* were significantly smaller (8–9 × 2–2.5 µm) than those of *S*. *myodes* [50], while bradyzoites of *S*. *montanaensis* (9.8–12.2 × 2.2–4.3 µm) were similar in size as compared to those detected in the present study [51]. Based on phylogenetic analyses, *S*. *myodes* is most closely related to *Sarcocystis* spp. that are known to employ predatory mammals as their definitive hosts, thus mammals rather than birds and reptiles are suggested definitive hosts of *S*. *myodes*. Comparative morphological analysis showed that *S*. *myodes* are similar to some *Sarcocystis* spp. utilizing reptiles or birds as their definitive hosts. Thus, *S*. *myodes* is different from other *Sarcocystis* spp. detected in voles based on morphological, genetic, and phylogenetic examination. The majority of *Sarcocystis* spp. described in voles are not strictly specific to one intermediate host species [1]. Therefore, it is likely that *S*. *myodes* can also be found in several vole species.

The most comprehensive morphological research on *Sarcocystis* spp. carried out in voles in Lithuania showed the presence of three morphological types of these parasites in the bank vole [39]. In the said study, 46 out of 320 examined bank voles were infected with *Sarcocystis*. Twenty *Sarcocystis* spp.-positive animals were chosen for morphological examination using LM. Sarcocystis were differentiated on the basis of the size of sarcocysts, the cyst wall appearance and morphometric parameters of bradyzoites. Sarcocysts similar to those detected in the present study were observed in the majority of tested animals (17/20). It should be pointed out that sarcocysts varied considerably in size (1.1–11.0 × 0.03–0.4 mm). Other two morphological types were characterized by sarcocysts having 3.2–4.2 µm densely packed hair-like protrusions and relatively small bradyzoites (7.0–9.2 × 1.8–2.5 µm) or sarcocysts with 2.7–4.0 µm bird claw-like protrusions and relatively large bradyzoites (11.2–13.0 × 2.5–4.2 µm). Thus, excluding *S*. *myodes*, at least two more *Sarcocystis* species should be present in the muscles of bank voles. In conclusion, many *Sarcocystis* spp. are described in rodents solely by morphological methods, and molecular revision of the already-named species is required, as the *Sarcocystis* group found in rodents is large and there is a general lack of recent research into these hosts.

## 5. Conclusions

From 45 bank voles examined, sarcocysts were discovered in a single specimen. Following investigation, a new species of *Sarcocystis* is described based on morphological LM and TEM analysis and five molecular loci, *18S* rDNA, *28S* rDNA, *cox1*, *rpoB*, and *ITS1* investigation. It is proposed to name the species “*S*. *myodes*”. Phylogenetic placement suggests mammals as the definitive hosts of *S*. *myodes*. Our findings indicate a lack of *Sarcocystis* studies in voles, as well as a gap in molecular data when studying *Sarcocystis* species whose intermediate hosts are rodents.

## Figures and Tables

**Figure 1 biology-11-00512-f001:**
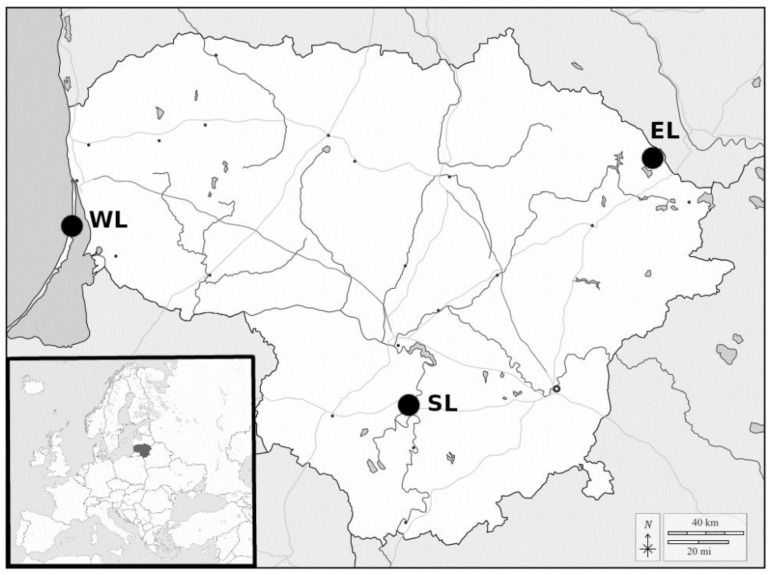
Investigation sites in Lithuania. The shore of the Curonian Lagoon in Western Lithuania (WL), the bank shore of the Nemunas River in Southern Lithuania (SL), the Peninsula of Lukštas Lake in Eastern Lithuania (EL).

**Figure 2 biology-11-00512-f002:**
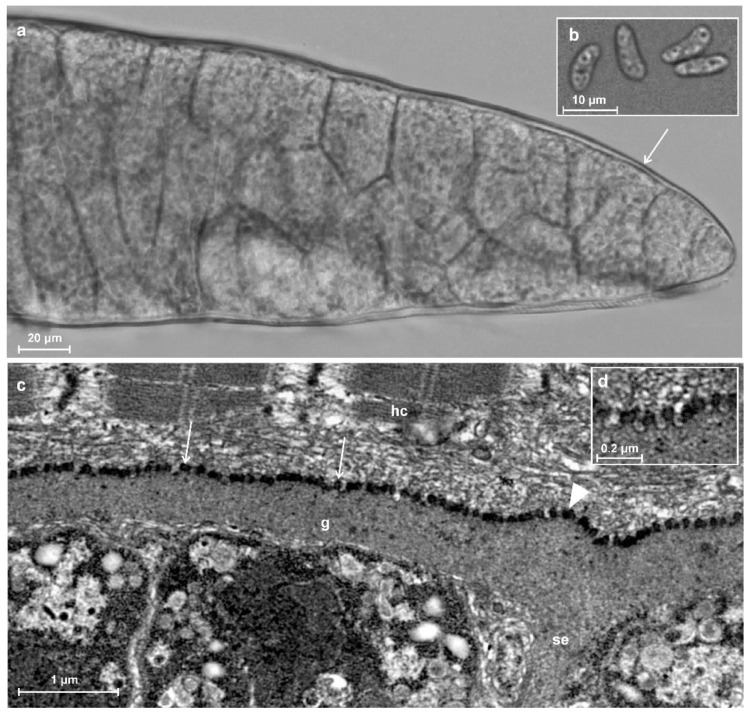
Morphological features of sarcocyst of *Sarcocystis myodes* isolated from skeletal muscles of the bank vole (*Clethrionomys glareolus*) from Lithuania. (**a**,**b**) LM analysis. Fresh muscle-squashed preparations. (**a**) Fragment of sarcocyst. Note a relatively thin and apparently smooth cyst wall (arrow). (**b**) Lancet-shaped bradyzoites. (**c**,**d**) TEM analysis. (**c**) A fragment of a straight cyst wall with a slight wave near septa (arrowhead); the parasitophorous vacuolar membrane has knob-like blebs (arrows). (**d**) Enlarged view on bleb-like structure on the sarcocyst wall; note muscular host cell (hc), septa (se), and ground substance (g).

**Figure 3 biology-11-00512-f003:**
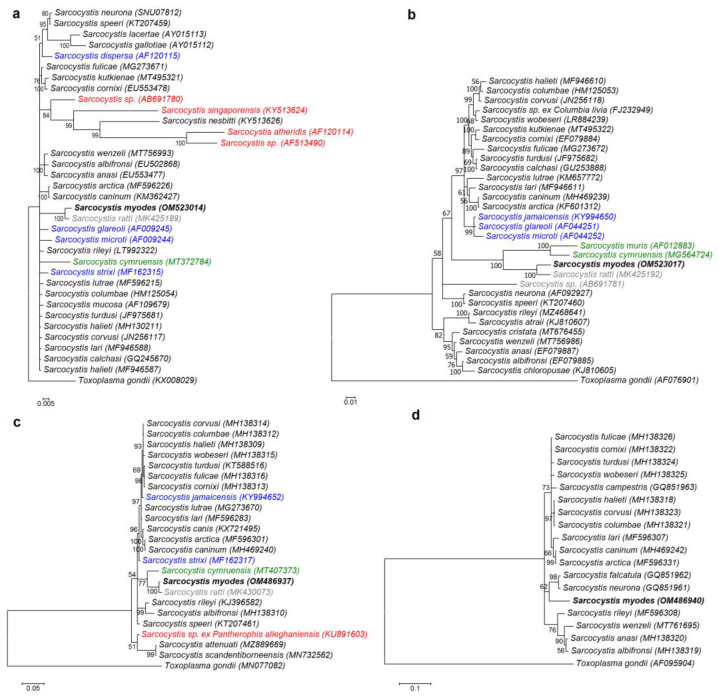
Phylogenetic trees of selected *Sarcocystis* species. The trees constructed based on (**a**) *18S* rDNA sequences, (**b**) *28S* rDNA sequences, (**c**) *cox1* sequences, and (**d**) *rpoB* sequences. The posterior probability support values are indicated next to branches. All color-coded *Sarcocystis* species are found in muscles of rodents. *Sarcocystis* species that are known to be distributed by birds are marked in blue, species that are distributed by snakes are marked in red, those transmitted by predatory mammals are in green, and species whose definitive hosts are not yet known are in a grey color.

**Table 1 biology-11-00512-t001:** Molecular characteristics of *Sarcocystis myodes* from bank vole.

Sequence Data	*18S* rDNA	*28S* rDNA	*cox1*	*rpoB*	*ITS1*
Sequence length (bp)	1786	1545	1053	762	969
Genbank acc. no.	OM523014–OM523016	OM523017–OM523019	OM486937–OM486939	OM486940–OM486942	OM523020–OM523022
Diversity amongst three isolates (%)	0	0	0	0	0–0.5
Highest sequences similarity values with other *Sarcocystis* spp.	*S*. *ratti* from black rat 99.5–99.6% (MT372787, MK425189); *S*. *rileyi* from mallard duck 98.1% (GU120092); *S*. *lutrae* from mustelids 98.0% (MG372102-03)	*S*. *ratti* from black rat 98.1% (MK425192); *S*. *muris* from house mouse 93.5% (AF012883); *S*. *cymruensis* from brown rat 93.5% (MG564724)	*S*. *ratti* from black rat 99.4% (MK430072, MT407374); *S*. *strixi* from laboratory mouse 95.9% (MF162317)	*S*. *fulicae* from Eurasian coot 95.7% (MH138326); *S*. *cornixi* from hooded crow 95.7% (MH138322), *S*. *caninum* from dog 95.7% (MH469242); *S*. *arctica* from red fox 95.4–95.7% (MF596311-31)	*S*. *ratti* from black rat 77.0–77.3% (MK910965)

## Data Availability

Not applicable.

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
