# Peer review of "Morphological and Molecular Description of Sarcocystis myodes n. sp. from the Bank Vole (Clethrionomys glareolus) in Lithuania"

_biology, 2022, doi:10.3390/biology11040512_

Round 1
Reviewer 1 Report
The work describes sarcocystis parasites found after rodent sampling in three different sites in Lithuania, in particular the bank vole Clethrionomys (Myodes) glareolus. 45 specimens were collected and only one sample was positive for the parasite. Morphological description and molecular analysis (on 5 markers) were provided to support the description of a new species of the genus.
The description of results and discussion are quite generic and must be implemented to support the description of the new species. No morphological comparison were provide with the closest phylogenetic species (S. ratti). I think that molecular distances of COX1 were not discussed adequately to support the distinguished species hypotesys.
Some suggestions follow:
Line 75, 81: Abbreviations/Initialisms should be defined the first time they appear
Line 99: you can add a short description to facilitate reading
Line 147: insert “on phylogenetic tree”
Line 136-154: 3.2. Molecular characterization and phylogeny:
In the paragraph much importance has been given to the description of the blast results (I suppose but there is no clear indication) and very little to the description of the results of the phylogenetic trees.
Moreover, there is a reason why there are not shown the ITS1 tree?
I suggest to perform the 5 markers combined phylogenetic tree, using different partition for each marker. The results could improve.
Table 1: Is the distance of cox1 enough to describe it as a new species? I think that is necessary to go deeper into the literature and discuss the use of DNA barcoding for this genus. I also suggest the use of some tools (eg. ASAP https://bioinfo.mnhn.fr/abi/public/asap/asapweb.html) or calculation of some more specific metrics (eg. evolutionary distance, nucleotide diversity, nucleotide divergence as in MEGA or DNAsp) to evaluate the actual genetic non-belonging to the S.ratti species.
Figure3: insert some indication in the figure, eg. hosts, highlight groups described in the text etc., it could improve the reading. Instead, all the details of the caption (alignment length, evolutionary model used etc.) should be put in the results.
Line 168: 3.3. Description of Sarcocystis myodes n. sp.: I think is important to clarify the morphological differences also with S. ratti, if any, since it is the most genetically similar species.
Line 175: how much is the genetic difference of the cox1 with these species?
Line 206-209: you can remove the fractions in brackets
Reviewer 2 Report
The rationale behind the research conducted by Eglė Rudaitytė-Lukošienė et al, I believe is well structured and conducted with scientific rigor. I have only a few things that I want to point out:
- In paragraph 2.3 of the materials and methods section, I believe they should create another sub-paragraph concerning the sequence analysis, in which the kit used to carry out the sequence amplification is specified (I suppose through the Sanger method).
- At the beginning of the results paragraph (line 114-115) a whole series of numbers and percentages are written in brackets which, I suppose, refer to the analysis conducted through the optical microscope (LM): please specify in more detail for those who are not very familiar with the technique!
- The points indicated in the arrows in figure 2 should be enlarged further. For example, you could use squares within the figure with less magnification.
- I believe the authors should show the agarose gel, which the amplifiers were loaded into and then sequenced.
- Finally, I was interested to know how you sequenced a fragment of about 1800bp, since in materials and methods it is said that the 18S rDNA fragment was amplified into two fragments of 900bp each, but in table 1 we refer to a single amplified one of 1786bp.
Round 2
Reviewer 1 Report
This manuscript has been well-improved by the authors after the implementation of most of the requested reviews. I support the manuscript for publication.